# The Mental Health and Wellbeing of University Students: Acceptability, Effectiveness, and Mechanisms of a Mindfulness-Based Course

**DOI:** 10.3390/ijerph18116023

**Published:** 2021-06-03

**Authors:** Emma Medlicott, Alice Phillips, Catherine Crane, Verena Hinze, Laura Taylor, Alice Tickell, Jesus Montero-Marin, Willem Kuyken

**Affiliations:** Department of Psychiatry, University of Oxford, Warneford Hospital, Oxford OX3 7JX, UK; emma.medlicott1@nhs.net (E.M.); arp91@bath.ac.uk (A.P.); catherine.crane@hmc.ox.ac.uk (C.C.); verena.hinze@psych.ox.ac.uk (V.H.); laura.taylor@hmc.ox.ac.uk (L.T.); alicetickell@hotmail.com (A.T.); jesus.monteromarin@psych.ox.ac.uk (J.M.-M.)

**Keywords:** mindfulness, university, student, mental health, wellbeing, resilience, self-compassion

## Abstract

Mental health problems are relatively common during university and adversely affect academic outcomes. Evidence suggests that mindfulness can support the mental health and wellbeing of university students. We explored the acceptability and effectiveness of an 8-week instructor-led mindfulness-based course (“Mindfulness: Finding Peace in a Frantic World”; Williams and Penman, 2011) on improving wellbeing and mental health (self-reported distress), orientation and motivation towards academic goals, and the mechanisms driving these changes. Eighty-six undergraduate and post-graduate students (>18 years) participated. Students engaged well with the course, with 36 (48.0%) completing the whole programme, 52 (69.3%) attending 7 out of 8 sessions, and 71 (94.7%) completing at least half. Significant improvements in wellbeing and mental health were found post-intervention and at 6-week follow-up. Improvements in wellbeing were mediated by mindfulness, self-compassion, and resilience. Improvements in mental health were mediated by improvements in mindfulness and resilience but not self-compassion. Significant improvements in students’ orientation to their academic goal, measured by “commitment” to, “likelihood” of achieving, and feeling more equipped with the “skills and resources” needed, were found at post-intervention and at 6-week follow-up. Whilst exploratory, the results suggest that this mindfulness intervention is acceptable and effective for university students and can support academic study.

## 1. Introduction

A growing proportion of the population is benefitting from the social, occupational, and academic opportunities offered by higher education. In recent years, there has been a three percent increase in higher education enrolments across both undergraduate and post-graduate university students, with larger numbers of ethnic minority students and students registering with a disability [1]. For many, higher education is a major transition met with increasing social, academic, and financial demands [2,3]. Rates of mental health difficulties amongst university students are noteworthy. A global survey of 13,984 students found 35% reported at least one DSM-IV mental disorder [4]. Rates of self-reported depression and anxiety in the United Kingdom are greater than the age-matched United Kingdom general population [5]. A growing body of research suggests that mental health difficulties worsen throughout the degree programme [6,7,8]. Whilst an increase in symptoms may not be caused by higher education itself, it is frequently suggested that the daily stressors associated with university life are a significant contributing factor [7]. Post-graduate students are burdened with additional daily stressors, including difficulties in the supervisory relationship and isolation [9].

The negative impact of having a mental health problem during university is broad, impacting the quality of life. Specifically, the presence of depressive symptoms in university students has been associated with role limitations due to physical health problems; while anxiety symptoms have been related to bodily pain; and both depressive and anxiety symptoms have been associated with reductions in general health, energy/fatigue, social functioning, as well as psychological distress, and lower psychological wellbeing [10]. It has also been observed that mental health problems during university affect academic performance [11] and the likelihood of dropping out [12]. Recently, UK Universities called for universities to transform institutions into “mentally healthy universities” that place the mental health and wellbeing of staff and students as foundational for all aspects of the university system [13]. The strategic framework developed by UK Universities outlines university-wide systematic changes which can be made across different domains (e.g., learning, support, etc.). For example, under the ‘learn’ domain, universities must make sure assessments “stretch and test” learning without imposing unnecessary stress. The ‘support’ domain focusses on implementing, in consultation with staff and students, safe and effective mental health interventions that are regularly audited for safety, quality and effectiveness [13]. Whilst cognitive, behavioural and mindfulness-based interventions have been found to effectively reduce symptoms of anxiety and depression in university students [14], the implementation of such interventions should be conducted in collaboration with university students to ensure that there is a demand for such programmes and that interventions are acceptable and accessible for all [13].

Over the last decades, there has been a growing interest both nationally and internationally in mindfulness-based programmes (MBPs) in higher education institutions [15]. Mindfulness is a natural, trainable capacity, which encourages people to approach experiences with attitudes of curiosity and care, and in ways that support overall wellbeing, and general functioning [16,17]. By training ‘mindful awareness’, MBPs aim to promote a conscious shift away from automatic, habitual responses that may increase distress towards greater self-regulation [17]. As well as cultivating mindfulness skills, the psycho-educational content within MBPs can be tailored to the target population whilst promoting an understanding of psychological processes at the core of distress and wellbeing [18]. A meta-analytic review of MBPs implemented in universities found overall improvements in depression, anxiety and wellbeing for students post-intervention, with lasting effects (>3 months) on distress [19]. MBPs should be further tested for their appropriateness with a student population, ensuring people across the spectrum of wellbeing can make use of them and at a level of intensity that balances accessibility with potency [20].

One example of an MBP is the “Mindfulness: Finding Peace in a Frantic World” course (M-FP) [21], which was adapted from Mindfulness-Based Cognitive Therapy (MBCT) [22] to be an accessible universal MBP. The course is developed for a non-clinical population and participants learn skills they can use in their daily lives to break the cycle of anxiety, stress, and exhaustion, as well as promote mental health and wellbeing [21]. M-FP themes include psycho-education, waking up from autopilot, how to respond to negative thoughts and difficult feelings, and self-care. Delivered in a university context, M-FP aims to support students to develop sustained awareness of the different aspects of daily living. Students are then encouraged to apply mindfulness skills to manage emotions, as well as academic and social pressures that may occur in day-to-day university life.

A self-guided version of the M-FP course in a university sample had positive effects on depression, anxiety, stress, satisfaction with life, mindfulness and self-compassion, with good acceptability [23]. A pragmatic randomised control trial using the instructor-led version of the M-FP course for university students alongside ‘mental health support as usual’ evidenced significant improvements in mental health difficulties in comparison to a control group [24].

Students also report both direct and indirect improvements in study skills and behaviour, including greater analytical thinking and memory capacity, increased enjoyment of studying, and reduced procrastination [25] following the M-FP course. Students reporting higher levels of mindfulness engage more with autonomous academic goals that are intrinsically motivated than students with lower levels of mindfulness [26], which is a consistent predictor of academic achievement across different educational contexts [27]. These findings are particularly relevant considering the disproportionate number of students with poor mental health who drop out of university or perform less well academically compared to those without mental health problems [11,12]. By providing students with internal resources to support wellbeing, mindfulness programmes could alleviate some of the disadvantages related to studying with poor mental health whilst potentially promoting positive academic behaviours. Trait mindfulness and behaving in accordance with intrinsic values are also related to greater wellbeing [28]. This is in line with Self-determination theory, which suggests that the relationship between mindfulness, intrinsic motivation and wellbeing is driven by greater awareness of self, leading to the choice of behaviours that are consistent with individual interests, values, and desires. It follows that awareness, cultivated through MBPs, may improve academic outcomes. Exploring the effects of the MF-P on students’ motivations towards academic goals is an aim of the current study.

Whilst the effects of MBPs on mental health and wellbeing are widely researched [29,30], the mechanisms through which these improvements are facilitated remain unclear. There is growing evidence to suggest that both mindfulness and self-compassion may be important mechanisms of change [31,32]. For example, in a study of an instructor-led M-FP tailored for secondary school teachers, there was reduced stress and increased rates of wellbeing, mindfulness and self-compassion [33]. Additionally, mindfulness and self-compassion have been shown to mediate improvements in wellbeing [34], stress, burnout and mental health [35].

The relationship between mindfulness and self-compassion is complex. Initially, it was conceptualised as a bidirectional relationship with each enhancing the other [36], but emerging research suggests that mindfulness and self-compassion improve mental health and wellbeing independently [37]. The delivery of the MBP may also impact these mechanisms for change, with the instructor-led delivery of the M-FP program indirectly improving wellbeing, mental health and burnout in secondary teachers by significantly enhancing mindfulness and self-compassion compared with the self-taught M-FP program [20]. Such mechanisms are also yet to be tested in a university student population.

Resilience is another key mechanism of change in MBPs [38]. Resilience refers to positive adaptation in the face of stress or trauma [39], a skill which is of particular importance to university students, who are faced with a number of novel challenging experiences (e.g., increased academic and financial demands). Drawing upon clinical models depicting mechanisms of change, the cultivation of mindfulness skills is thought to enhance self-regulation skills, promoting the psychological resilience that supports mental health and wellbeing [40]. This is supported by cross-sectional and experimental research, revealing a partially mediating serial relationship between mindfulness, resilience and subjective wellbeing in community, and university samples [38,41]. In a recent cross-sectional study of general population participants, significant direct effects of mindfulness, self-compassion and resilience on anxiety and depression symptoms were observed, and also indirect effects of mindfulness and self-compassion through resilience on depression symptoms were found [42].

An improved understanding of the mechanisms of change for MBPs would enable further refinement of these programmes, thereby optimising individual outcomes. Mindfulness, self-compassion, and resilience are likely to be important mechanisms in MBPs for improving wellbeing and mental health outcomes in students. Based on the literature presented, Figure 1 shows a proposed model whereby there is a sequential relationship between mindfulness, self-compassion, resilience, mental health, and academic outcomes. Testing this model is beyond the analytic and methodological capacity of this paper but forms the theoretical basis of the analyses conducted.

### Study Aims

The present study explores engagement with acceptability and effectiveness of the instructor-led M-FP course [21] in improving mental health and wellbeing in university students. Second, the impact of the M-FP course on the students’ orientation and motivation towards their academic goals is also explored. Finally, three mechanisms of change were independently explored in relation to mental health and wellbeing: (1) mindfulness, (2) self-compassion, and (3) resilience.

## 2. Materials and Methods

### 2.1. Study Design

The study was conducted at the University of Oxford and utilised an open pre-post-test intervention design with a 6-week post-course follow-up. Participants gave informed consent and completed self-report questionnaires online of the outcomes and mechanisms. The questionnaires were administered using Qualtrics software (Qualtrics, Provo, UT, USA). Acceptability was assessed through both bespoke questions as well as engagement with the programme monitored by the M-FP teachers.

### 2.2. Participants and Procedure

Participants were undergraduate and post-graduate students (aged ≥18 years) who signed up to an 8-week group-based mindfulness course offered by the Oxford Mindfulness Centre and volunteered to participate in the study. Prior to course enrolment, participants self-assessed the suitability of the mindfulness course based on information supplied by the Oxford Mindfulness Centre [43], including a statement on contraindications (such as serious mental or physical health concerns or recent bereavement) to the mindfulness course. Expression of interest for the present study occurred between October 2017 and April 2019, and therefore participants were consecutively recruited during this time. The research team contacted those who expressed an interest, provided them with information about the study and an opportunity to ask questions. Participants were then sent a link to consent to the study via an online form. During the study period, 200 and 96 students took part across 10 mindfulness groups, each group following the 8-week mindfulness-based programme. Of these, 86 students (29%) consented to participate in the study.

Data was collected pre-intervention, one week prior to the first-course session (T0), post-intervention, at the end of the 8-week course (T1), and at a 6-week follow-up after completion (T2). Participants received £20 after completing the first 2 time points and an additional £10 for continued participation at T2 (a maximum of £30 in total) as a reimbursement for their time. The payment was not conditional on attendance at the mindfulness course.

A risk and safeguarding protocol was implemented to protect participants’ safety. Those reporting significant distress were provided with information on sources of support. Additionally, participants who reported suicidal ideation were contacted for further assessment and, when appropriate, the student’s college nurse was informed. The University of Oxford Research Ethics Committee approved the study (R52786/RE004).

#### Mindfulness-Based Programme

The MBP used for the intervention was the M-FP [21]. The course is an adaptation of Mindfulness-Based Cognitive Therapy (MBCT) [44] and includes 8 in-person weekly 90-min sessions of reduced-intensity MBCT with a focus on reducing general distress and improving wellbeing [21]. The main themes include “waking up to the autopilot,” “keeping the body in mind,” “the mouse in the maze,” “moving beyond the rumour mill,” “turning towards difficulties—from reacting to responding,” “practicing kindness,” “when did you stop dancing?” and “your wild and precious life.” Students paid £65 to take part in the MBP, which was the standard price of the course, a cost unrelated to the study. Each of the 10 mindfulness groups that did the 8-week programme consisted of up to 30 participants (M = 29.6, SD = 0.97). Students were recommended to engage in daily home practice from the course books audio files and activities, such as mindful walking and eating, amongst others. All groups were taught by a qualified mindfulness teacher from the Oxford Mindfulness Centre, who met the good practice guidelines developed by UK Network for Mindfulness-Based Teacher Training Organizations (http://mindfulnessteachersuk.org.uk/#guidelines, accessed on 2 June 2021). The mindfulness teachers were not involved in the research study and were unaware of which students were study participants.

### 2.3. Measures

#### 2.3.1. Demographic Information

At baseline (T0), participants were asked to provide information on their age, gender (Male, Female, Other, Prefer not to say), ethnicity (White British; White Irish; Any other White background; White and Black Caribbean; White and Black African; White and Asian; Any other mixed/multiple ethnic backgrounds; Indian; Pakistani; Bangladeshi; Chinese; Any other Asian background; Caribbean; African; Any other Black/African/Caribbean background; Arab; Any other ethnic origin group; Prefer not to say) and education (Bachelor’s degree, Master’s degree, Doctorate degree, Other). Participants were additionally asked whether they were currently suffering from a mental health disorder or had previously received a diagnosis of a mental health disorder and were asked to specify their diagnosis. Furthermore, we collected information on the students’ previous experiences with meditation in the format of an open answer question.

#### 2.3.2. Expectation of Benefit, Engagement and Programme Acceptability

At T0, participants were asked to report how much they expected to benefit from the mindfulness course on a Likert scale [‘not at all’ (0) to ‘very much’ (10)]. Following the mindfulness intervention (T1), participants were asked to report on the number of mindfulness sessions they had attended (0–8) and the number of days and minutes they had practiced mindfulness at home during the eight-week course. Course acceptability was assessed using four statements, which the participants rated from 0 to 10: “How much do you feel that you benefitted from the course?”; “Please rate the quality of teaching,” “Mindfulness courses should be made widely available to students at the University of Oxford,” and “How likely are you to use mindfulness in the future?” A mean of these four statements was used to calculate a “total acceptability score.” The internal consistency of these four statements was good in the current sample (alpha = 0.78).

#### 2.3.3. Wellbeing

At all three time points (T0, T1 and T2), the Warwick-Edinburgh Mental Wellbeing Scale (WEMWBS) [45] was administered. The WEMWBS is a psychometrically robust 14-item measure, assessing wellbeing over the past two weeks. Upon being presented with a statement (e.g., I have been feeling good about myself), participants were asked to score each item on a 5-point scale [“none of the time” (1) to “all of the time” (5)], yielding a total score ranging from 14 to 70. A higher total score indicated greater wellbeing [45]. Internal consistency in the current sample was good (T0 alpha = 0.86; T1 alpha = 0.87; T2 alpha = 0.90).

#### 2.3.4. Mental Health

The Clinical Outcomes Routine Evaluation-10 (CORE-10) [46] is derived from the 34-item Clinical Outcomes Routine Evaluation-Outcome Measure [47] and was used as a measure of psychological distress at all data collection time points (T0, T1 and T2). This questionnaire consists of 10 statements about thoughts and feelings (e.g., “I have felt tense, anxious, or nervous”). Participants were asked how often they felt this way over the past week, using a 5-point Likert-type scale, [“not at all” (0) to “most or all of the time” (4)]. The total score ranges from 0 to 40; a score of 11 or higher is considered clinically significant. The CORE-10 has shown good validity, internal consistency, and sensitivity to change [46]. The internal consistency in the current sample was good (T0: alpha = 0.84, T1: alpha = 0.81, T2: alpha = 0.82).

#### 2.3.5. Mindfulness

At all data collection time points (T0, T1 and T2), participants completed the Five-Facet Mindfulness Questionnaire-Short Form (FFMQ-SF) [48], which is a shortened, 15-item version of the Five-Facet Mindfulness Questionnaire (FFMQ) [49]. Participants rated statements (e.g., “I’m good at finding words to describe my feelings”) on a 5-point scale [“never or rarely true” (1) to “very often or always true” (5)]. As per recommendations for pre–post research [50], the “observation” subscale was excluded from the total score calculations, leading to a range of 12–60. The internal consistency in the current sample was appropriate (T0: alpha = 0.81, T1: alpha = 0.82, T2: alpha = 0.75).

#### 2.3.6. Self-Compassion

The present study used a short-form version of the Self-Compassion Scale (SCS-SF) [51] to measure self-compassion at all 3 time points (T0, T1 and T2). With 12 items rather than 26 [51,52], this measure has good convergent validity and reliability [51]. Participants were presented with a statement (e.g., “When I’m going through a very hard time, I give myself the caring and tenderness I need”), which they rated in terms of the frequency of this experience, using a five-point scale [“almost never” (1) to “almost always” (5)]. We used a total score of self-compassion [53] that ranged from 1 to 5, with higher values indicating greater levels of self-compassion. Internal consistency in the current sample was good (T0 alpha = 0.84; T1 alpha = 0.81; T2 alpha = 0.82).

#### 2.3.7. Resilience

The Connor–Davidson Resilience Scale (CD-RISC) [54] is a 25-item measure of resilience and coping. At all three time points (T0, T1 and T2), resilience was assessed using a shorter 10-item version (CD-RISC-10) [39], which has a stable factor structure, good reliability and validity [39]. Participants rated each item (e.g., “I believe I can achieve my goals, even if there are obstacles”) for their “truthfulness” over the past month on a 5-point scale [‘not true at all’ (0) to ‘true nearly all the time’ (4)]. The total score can range between 0 and 40, with a higher score indicating greater resilience [39]. Internal consistency in the current sample was good (T0: alpha = 0.88, T1: alpha = 0.89, T2: alpha = 0.90).

#### 2.3.8. Academic Goals

The ‘Measure to elicit positive future goals and plans’ (MEPGAP) [55] has been adapted in previous research to measure conditional goal setting [56,57]. For the purpose of the present study, we further adapted this measure to assess students’ academic goals. At each time point, participants were asked to generate their most important “academic goal,” defined as “something that they would like to happen or to be true of their academic life in the future.” Subsequently, participants were asked to rate (1) the likelihood of achieving this goal, (2) the extent to which they felt that they have the skills and resources necessary to obtain this goal and (3) how committed do they felt to attaining this goal on a scale of 0–10. Intrinsic and extrinsic motivation towards achieving this goal was measured by asking participants to rate the following two statements (1) “I am pursuing this goal because someone else wants me to, or because I will get something from somebody if I do. I probably wouldn’t pursue it if I didn’t get some kind of reward, praise, or approval for it” and (2) “I am pursuing this goal because I really believe that it is an important goal I have. I endorse it freely and value it wholeheartedly.” from (“not at all because of this reason” 0 to “completely because of this reason” 10). Each subscale (i.e., likelihood, skills and resources, commitment, intrinsic motivation and extrinsic motivation) was treated as a separate measure of academic goal pursuit, with a higher subscale score indicating greater orientation towards academic goals on that specific domain.

### 2.4. Data Analyses

#### 2.4.1. Participant Characteristics

Baseline socio-demographic characteristics were described using means (SDs) for continuous data and frequencies (percentages) for categorical data. Associations between socio-demographic characteristics and research attrition were analysed at post-test and follow-up, using odds ratios obtained from bivariate logistic regression analyses.

#### 2.4.2. Expectations of Benefit, Engagement in and Acceptability of the Course

We used the mean (SD) to describe expectations of benefit from the mindfulness course. The engagement was presented in terms of the number of sessions attended, using frequencies (percentages), and the rates of home practice during the programme, using medians and interquartile ranges (IQRs). Participant ratings of acceptability were described using means (SDs). Using bivariate linear regression analyses and standardised beta coefficients (β), we explored the possible relationships between baseline characteristics and the number of sessions attended, as well as between course acceptability and pre–post differential scores in outcomes.

#### 2.4.3. Effectiveness of the Mindfulness Intervention

The primary analysis was carried out for the main outcomes of wellbeing and mental health at post-test (T1) and follow-up (T2), using intention-to-treat analysis (ITT). We used mixed linear regression analyses in which time was entered as the independent variable, and within-person variance was captured by the random part of the model. We used the restricted maximum likelihood algorithm (REML) to obtain less biased estimates of parameters given the small sample size [58]. We calculated unstandardised regression coefficients (B) from complete cases for our primary analyses. Secondarily, we developed sensitivity analyses carrying out adjusted—controlling for previous experience of mindfulness or meditation and expectancy at baseline—and imputed models. Imputed models used linear multiple imputations of 20 datasets based on chained equations to address missing data at post-test and follow-up in the main outcomes (i.e., distress and wellbeing), considering: (a) those variables included in the primary outcome analysis; (b) variables significantly related to, or potentially related to, non-response (see Appendix A); (c) variables that explained a significant amount of variance in the main outcomes (see Appendix A for a list of the included variables). Additionally, we calculated the possible “time × current diagnosis of mental health problems” and the “time × previous diagnosis of mental health problems” interactions on distress. We also estimated the effect of time on the mechanistic variables of mindfulness, self-compassion, and resilience, and the effect of time on the secondary outcomes (i.e., the student’s orientation towards their academic goals). We reported within-group effect sizes (ESs) from the marginal means by correcting for the dependence of the repeated measures [59]. For the proposed interactions, we computed ESs for pairwise comparisons, using the pooled pre-test standard deviation to weight the differences in the pre‒post marginal means and to correct for the population estimate [60]. Standardized ESs of d ≤ 0.20 are considered small, d = 0.50 as medium, and d ≥ 0.80 or more as large [61].

#### 2.4.4. Engagement during the Programme and Outcomes 

Using Spearman’s correlations, we explored the possible relationships between the number of mindfulness sessions attended as well as the amount of practice during the programme—calculated as the multiplication of the number of days practiced and the estimated mean practice duration per day—and the pre–post changes in each outcome. Additionally, we also completed these same analyses on a subset of participants with current mental health difficulties.

#### 2.4.5. Mediating Role of Mindfulness, Self-Compassion and Resilience

To test the possible mediating effects of mindfulness, self-compassion, and resilience, we used an exploratory within-participant path-analytic framework [62,63]. Firstly, we developed a post hoc power analysis for participants with complete data. Assuming a partial mediation scenario, we estimated the product of the tests of paths “a” and “b,” which closely estimates the power of bootstrap estimates [64,65] for a large effect in both paths “a” and “b” (i.e., a standardised path-value of 0.40 each) and an intermediate effect in path “c’” (i.e., the direct effect after conditioning on the indirect effects, with a path-value of around 0.30) of the mediation models. Secondly, we evaluated the correlations between pre–post-intervention changes in mindfulness, self-compassion, and resilience, with pre-intervention to follow-up changes in the primary outcomes of wellbeing and distress. Finally, we explored the relationships between time (i.e., the independent variable), the pre‒post-intervention changes in the mediators (i.e., mindfulness, self-compassion, and resilience), and the pre-intervention to follow-up changes in the main outcomes (i.e., distress and wellbeing), using ordinary least squares (OLS) analyses. For these analyses, we used unstandardised path estimates from the regression coefficients and developed independent models for each tested mediator (i.e., mindfulness, self-compassion, and resilience) and main outcome (i.e., distress and wellbeing). Thus, we entered ‘time’ as the independent variable (X), the mindfulness (or self-compassion or resilience) pre‒post-intervention change scores as a possible mediator (M), and the pre-intervention to follow-up change scores for wellbeing (or distress) as a dependent variable (Y). The simple within-group path mediational model followed is graphically depicted in Figure 2. We calculated the unstandardised regression coefficients and the corresponding standard errors for bootstrapped indirect effects. We present the 95% bias-corrected confidence interval based on 10,000 bootstrap samples to overcome possible problems of asymmetry in the distribution of the indirect effects. Indirect effects are considered statistically significant when their 95% confidence interval does not include zero [66]. Finally, we used the multiple determination coefficient (R^2^) to calculate the ESs for the mediation models (i.e., values of 0.00 = null effect, 0.14 = small effects, 0.39 = medium effects, and 0.59 or more = large effects) [67].

An overall 2-sided α level of 0.05 was used. We did not correct for multiple testing, given the exploratory nature of the present study [68]. Analyses were performed using STATA v17.0 (StataCorp. College Station, TX, USA) and IBM SPSS v27.0 (IBM Corp. Armonk, NY, USA).

## 3. Results

### 3.1. Participant Characteristics and Participant Flow

Table 1 provides an overview of the participant characteristics at baseline, and a flowchart of participants can be seen in Figure 3. Research attrition was predicted by age, the current presence of a diagnosed mental health problem, as well as baseline levels of mindfulness, self-compassion, distress, and the secondary outcome of commitment towards academic goals (see Appendix A). No other variables were involved in the missing data pattern.

### 3.2. Expectations of Benefit, Engagement in and Acceptability of the Course

The expectation of benefit was M = 7.09 (SD = 1.64). Of the 74 participants who completed T1 measures, 36 (48.0%) participants attended the whole M-FP programme, and including these, 52 (69.3%) attended seven out of eight sessions, with 71 (94.7%) attending half or more of the programme (Figure 3). The median number of days participants reported practising mindfulness meditation during the programme was 5 days per week (IQR: 3–7), and on these days, they reported engaging in mindfulness practices for a median of 10–15 min per day (IQR: 5–10 to 15–20). Participant ratings of acceptability, following the intervention, were as follows: “the intervention was beneficial”, M = 6.93 (SD = 1.90); “the intervention should be made widely available”, M = 9.12 (SD = 1.21); “quality level of the teaching,” M = 8.05 (SD = 1.97); “likelihood of using mindfulness in the future,” M = 8.12 (SD = 1.99). The total acceptability score had a mean value of M = 8.06 (SD = 1.40). Expectations of benefit from the mindfulness course was not a predictor of completion of the programme (β = 0.17; *p* = 0.148). However, participants classified as ‘White,’ including ‘White British, ‘White Irish, any ‘Other White’ ethnicity, were significantly associated with a higher number of sessions attended compared with other ethnic groups (β = 0.28; *p* = 0.019), and thus, ethnicity was a predictor of completion of the programme. 

The expectations of benefit from the mindfulness course was significantly related to the total acceptability score (β = 0.50; *p* < 0.001), while the total acceptability score was significantly related to pre‒post improvements in mindfulness (β = 0.29; *p* = 0.012), self-compassion (β = 0.25; *p* = 0.030), resilience (β = 0.39; *p* = 0.001) and wellbeing (β = 0.40; *p* < 0.001). No other significant relationships were found between expectations of benefit, total acceptability score and pre–post differential scores in other outcomes.

### 3.3. Effectiveness of the M-FP Programme

Table 2 shows that there were significant improvements in the main outcomes of wellbeing and distress, following the intervention at T1 (WEMWBS: B = 2.08; *p* < 0.001; CORE: B = −2.63; *p* < 0.001) with moderate effects, and at T2 (WEMWBS: B = 2.09; *p* < 0.001; CORE: B = −2.39; *p* = 0.001), with moderate and low-moderate effects, respectively. Adjusted models and analyses using imputed data reinforced these results, with significant but reduced slopes (see Appendix A).

As can be seen in Table 3, there was an interaction effect between time and current mental health problems on distress at post-intervention and follow-up, with large effects. Furthermore, the analyses revealed a significant interaction effect between time and previously diagnosed mental health problems on distress at post-intervention and follow-up, with moderate to large effects. Those participants suffering from a current or previous history of mental health problems showed significantly more improvements in distress than those without a current or a previous history of mental health problems. These results were replicated when using adjusted models and imputations, although ESs were reduced when using imputations (see Appendix A).

There were significant pre–post-intervention and pre-intervention to follow-up improvements in the mediators, including mindfulness, self-compassion, and resilience, with moderate effects (Table 4). Furthermore, the analyses revealed significant pre−post-intervention and pre-intervention to follow-up improvements in the secondary outcomes, including the domains of likelihood, skills and resources, and commitment towards achieving one’s academic goals, with moderate to low effects. Intrinsic and extrinsic motivation did not show significant changes at any time point (Table 5).

### 3.4. Engagement during the Programme and Outcomes

The number of mindfulness sessions attended during the programme was not significantly related to improvements between time points across all measures. However, the amount of home practice during the programme (calculated as the multiplication of the number of practice days and the estimated mean time practiced each day) was significantly related to pre‒post-intervention improvements in wellbeing (R = 0.28; *p* = 0.017), mindfulness (R = 0.23; *p* = 0.045) and self-compassion (R = 0.27; *p* = 0.021). Although the amount of home practice during the programme was not significantly related to pre‒post-intervention improvements in distress (R = −0.04; *p* = 0.716) and resilience (R = 0.12; *p* = 0.327), intermediate effects were observed in a sub-group of students, who reported current mental health problems (n = 16; distress: R = −0.36, *p* = 0.169; resilience: R = 0.33, *p* = 0.207).

### 3.5. Mediating Role of Mindfulness, Self-Compassion and Resilience

Firstly, we computed bivariate correlations between pre‒post-intervention differences in mindfulness, self-compassion, and resilience and pre-intervention to follow-up differences in wellbeing, and distress (see Appendix A). The shared variance between pre‒post-intervention changes in mindfulness and self-compassion was 36%; between mindfulness and resilience, the shared variance was 30%; between self-compassion and resilience, it was 19%, and it was 29% between pre-intervention to follow-up changes in wellbeing and distress. The path analysis results are presented in Table 6, Table 7 and Table 8 and illustrated in Figure 2.

We observed that participants reported significant improvements in mindfulness at post-intervention (a = 3.80; *p* < 0.001) and that these improvements predicted changes in wellbeing (b = 0.29; *p* < 0.001) and distress (b = −0.42; *p* < 0.001) at follow-up (see Table 6). The 95% bias-corrected bootstrap confidence intervals for the interaction effects on wellbeing (0.38−2.05) and distress (−2.65 to −0.69) did not cross zero, indicating a possible mediating effect of mindfulness on both wellbeing and distress. This mediating effect explained 56% of total effects in wellbeing and 75% of total effects in distress, with small to medium ESs (Table 6). Furthermore, participants reported significant improvements in self-compassion at post-intervention (a = 0.38; *p* < 0.001), and these changes predicted improvements in wellbeing (b = 2.57; *p* = 0.005) at follow up (see Table 7). The 95% bias-corrected bootstrap confidence intervals for the interaction effects on wellbeing (0.22 to 1.98) did not cross zero, indicating a possible mediating effect of self-compassion on wellbeing, explaining a 51% of total effects, with a small ES. The possible mediating effect of self-compassion on distress was not significant (Table 7). Finally, participants reported significant improvements in resilience following the intervention (a = 2.07; *p* = 0.004), and these improvements predicted changes in both wellbeing (b = 0.25; *p* = 0.013) and distress (b = −0.33; *p* = 0.033) at follow−up. The 95% bias-corrected bootstrap confidence interval for the interaction effects on wellbeing (0.12–1.10) and distress (−1.59 to −0.07) did not cross zero, indicating a possible mediating effect of resilience on wellbeing and distress, explaining a 27% and 33% of total effects respectively, with small ESs (Table 8).

## 4. Discussion

This study explored the acceptability and effectiveness of the ‘Mindfulness: Finding Peace in a Frantic World’ course [21] in a university student population. Exploratory investigations were also conducted into changes in attitudes and motivation towards a self-selected academic goal. Preliminary explorations were conducted into the mediating role of mindfulness, self-compassion, and resilience for wellbeing and mental health (self-reported distress) independently. Results suggest that university students engaged with the M-FP course and reported high rates of acceptability. In addition, there were significant improvements in wellbeing and decreased mental health difficulties following the intervention and at 6-week follow-up, despite just under half of the sample attending all eight sessions. Improvements in wellbeing were significantly mediated by mindfulness, self-compassion, and resilience, whilst reductions in mental health problems were mediated by improvements in mindfulness and resilience, but not self-compassion. Further exploration revealed significant improvement in perceived “commitment” to, “likelihood” of achieving, and feeling more equipped with the “skills and resources” required to accomplish a self-selected academic goal at post-intervention and at 6-week follow-up. No improvements were revealed for intrinsic or extrinsic motivation towards academic goals. 

The M-FP course had a moderate effect in improving wellbeing and decreasing mental health problems at post-intervention, with moderate-low effects at 6-week follow-up, which reflects similar findings from recent meta-analyses and systematic reviews exploring university student samples [19,69]. Consistent with previous research, improvements in wellbeing were associated with more time engaging in home practices [70]. Engagement, by M-FP course attendance, was higher than alternative MBPs offered to university students [23], and slightly lower in comparison to a secondary teacher sample [20]. Engagement by home practice was lower than the recommended practice duration within the 8-week M-FP course (20 min per day) [21], but it was within the previously reported total range of home practice for MBPs in university student samples [23,24]. Thus, universities might be implicated in improving student well-being by enhancing mindfulness home practice (e.g., making available tools that facilitate practice such as online platforms with tutorials, audios, videos, etc.). Participants perceived the course to be beneficial and useful, and many had intentions to continue using mindfulness in the future. 

Ethnicity was a significant predictor of completion of the programme, and those who categorised themselves as being of white ethnicity attended significantly more sessions than other ethnic groups, although we recognise this is a small sample (Table 1). This raises an important perspective on the inclusivity and accessibility of mindfulness interventions for individuals from minority ethnic backgrounds. Similar studies in the area [23,24] have not reported the relationship between ethnicity and attendance but have similar percentages of ethnic diversity in their participant groups. In a systematic review of Mindfulness and Meditation-Based Interventions (MMBI), only 24 out of 12,265 studies were identified as ‘diversity-focused’ [71]. Research efforts, therefore, need to be made to ensure such courses are accessible, acceptable and effective to people across a range of ethnic backgrounds at the stage of both design and delivery.

The present student sample reported moderate (26%) rates of current mental health problems in line with recent student population estimates [4,10,72]. Commonly, mindfulness practitioners advise against participation in MBPs if a potential participant is experiencing severe psychological distress, current depression, mania, or recent bereavement. In the present study, participants who reported experiencing current or past mental health problems reported significantly greater reductions in distress compared to those without current or past mental health problems. This reflects similar evidence from Galante et al. [23], which found that baseline levels of distress moderated the benefit obtained from an MBP in a university student population. These findings could have important implications for the implementation and benefits of MBPs with vulnerable students and should be explored in further research.

There is preliminary evidence to suggest that MBPs implemented in a student population can impact academic outcomes and behaviour [23,25]. Whilst academic outcomes were not measured in the present study, we found an effect of time on students’ orientation towards their most important academic goal, suggesting the M-FP may have facilitated a more positive orientation towards academic goals. Intrinsic motivation is widely considered an important determinant for academic success [27] and is associated with trait mindfulness [73]. However, the present study did not find any significant changes in intrinsic and extrinsic motivation towards obtaining an academic goal following participation in the M-FP course. This finding is likely the result of ceiling effects, as much of the study sample reported very high levels of intrinsic motivation and very low levels of extrinsic motivation at baseline. This may arguably be characteristic of the present sample, comprising of students who have met the high entry requirements for studying at Oxford University [74]. Future research would benefit from a more diverse sample representative of a broad university student population. However, this result could also be derived from the instrument that was used, which only included one single item to measure each academic goal domain. Nevertheless, these preliminary findings contribute to our understanding of the effectiveness of the M-FP course on improving students’ orientation towards their academic goals and may form the basis of future investigations.

In line with systematic reviews and meta-analyses exploring mechanisms of change in MBPs [31,32], mindfulness, resilience, and self-compassion were found to be significant mediators for pre-post-intervention changes in wellbeing. Furthermore, whilst improvements in distress were mediated by resilience and mindfulness, self-compassion was not found to significantly mediate improvements in distress in the present sample. This finding is consistent with previous findings from a study with a sample of secondary school teachers that found mindfulness, but not self-compassion, mediated changes between the frequency of mindfulness practice during an MBP and mental health outcomes [20], suggesting that improvement in mental health symptoms and wellbeing may be driven by different pathways of change. Recent cross-sectional evidence from general population participants also found significant direct effects of mindfulness, self-compassion, and resilience on anxiety and depression symptoms, with indirect effects of mindfulness and self-compassion through resilience on depression symptoms [42]. Thus, in order to optimise outcomes and delivery of MBPs, the disparity between mechanisms underpinning improvements in wellbeing and mental health symptoms, including distress, should be investigated further.

The findings are interpreted in the light of several limitations. Given the exploratory nature of our study, we did not control for multiple comparisons, possibly leading to false-positive findings. The study is characterised by a small sample size and large rates of attrition. Although statistical techniques were employed to address these characteristics, it may not have sufficient power to fully investigate all the questions proposed with the potential to lead to false-negative findings. Therefore, it is important that these findings are considered preliminary and that future research aims to formally test these observations in larger university student samples. The absence of a control condition means that it is unclear whether improvements in wellbeing, distress, mediators and the students’ orientation towards their academic goals can be attributed to their participation in the M-FP course or the passage of time. Students’ orientation towards their academic goals is a new line of enquiry. Thus, it is currently unknown how academic goal orientation (i.e., commitment to, likelihood of achieving, and having the skills and resources to achieve the academic goal) may fluctuate during the academic year. The present study did not explore the mechanisms through which the observed changes in academic goal orientation were facilitated to avoid over-testing. Future research may wish to explore these longitudinal changes by combining self-report measures of academic goal orientation with objective measures of academic achievement and study behaviour. As mentioned, the academic goal findings may also be derived from the instrument that was used, which only included one single item to measure each academic goal domain, and future research should improve the quality of the measures used in this regard. In addition, whilst the M-FP course was found to be particularly effective for students with mental health problems, these findings can also be explained by the observation of regression to the mean, as students reported high levels of mental health problems at study entry [75]. Finally, opportunity sampling was employed, whereby students, who were already considering partaking in the M-FP course, were recruited. Hence, participants may present with characteristics (e.g., financial resources to afford course participation), which differ from the general student population at large, limiting the generalisability of the present findings. In this sense, we observed that ethnicity predicted completion of the programme, and that raises questions of inclusivity. For example, marginalized identities (rather racial, gender, gender identity, sexual orientation, migrant, etc.) may play a role in the outcomes and would require special consideration, as findings may be differential based on students’ identity.

## 5. Conclusions

Whilst the findings are exploratory in nature and must be considered preliminary, they highlight possible avenues for future investigations. Considering the limitations, the M-FP course was found to be acceptable and effective in the university student population. Students who undertook the course showed improvements in wellbeing and distress over the study period. Whilst this may also be the result of regression to the mean, mindfulness, self-compassion, and resilience were found to mediate changes in wellbeing, while changes in distress were mediated by mindfulness and resilience. Such mediation effects would be expected of active intervention. Further, to our knowledge, this study is the first to suggest that participation in the M-FP course may improve students’ orientation towards their academic goals (i.e., the perceived likelihood, commitment, and skills and resources of achieving their goal). Given the exploratory nature of this study, future research should aim to formally test these observations in larger student samples, using randomised controlled trial designs and combining more robust self-report measures of academic goal orientation with objective measures of academic achievement, and study behaviour. In light of the corresponding relationship between ethnicity and completion of the intervention, it is pertinent that barriers to attendance and engagement for students from ethnic minority backgrounds are explored. To optimise outcomes of the M-FP course for university students, the disparity between mechanisms underpinning improvements in wellbeing and distress should be investigated further. The results merit future investigation whilst having implications for public health and calls for universities to further support and promote positive mental health in their students.

## Figures and Tables

**Figure 1 ijerph-18-06023-f001:**
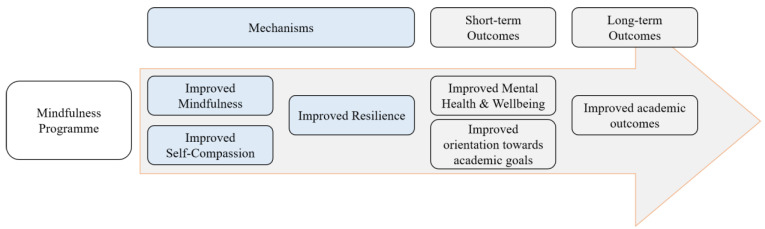
The proposed model of change following participation in a MBP for university students.

**Figure 2 ijerph-18-06023-f002:**
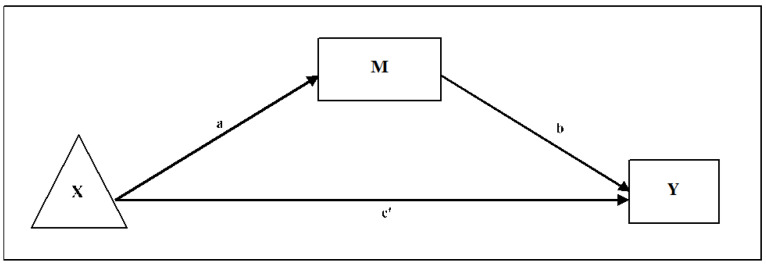
The independent variable is the repeated-measures factor (e.g., time: X). M is the pre‒post difference in the corresponding mechanistic variable. The dependent variable is the pre-follow-up difference in the corresponding main outcome (Y). “a*b” = indirect effect through the mediator. “c’” = direct effect after adjusting for the mediating effects.

**Figure 3 ijerph-18-06023-f003:**
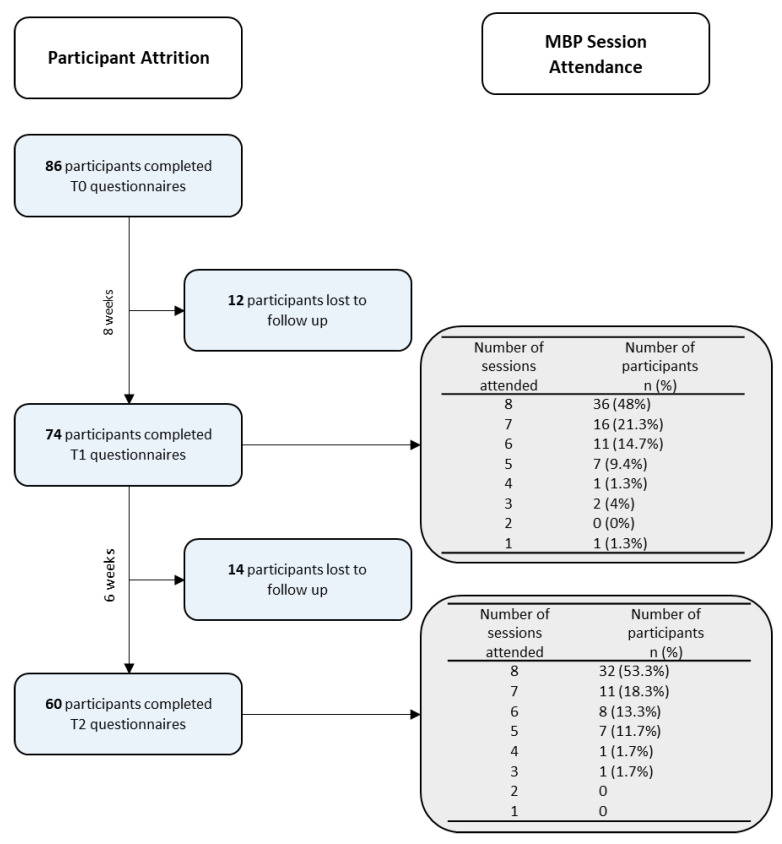
Flowchart of participants.

**Table 1 ijerph-18-06023-t001:** Baseline characteristics of the study sample.

Demographics	Sample (n = 86)
Gender, n (%)	
Female	58 (67%)
Male	27 (31.4%)
Other	1 (1.2%)
Age, Mn (SD)	24.91 (6.33)
Ethnicity, n (%)	
Arab	1 (1.2%)
Chinese	4 (4.7%)
Indian	3 (3.5%)
Other Asian	2 (2.3%)
Other Mixed/Multiple Ethnicities	3 (3.5%)
Other White	27 (31.4%)
Prefer not to say	3 (3.5%)
White Asian	4 (4.7%)
White British	37 (43.0%)
White Irish	2 (2.3%)
Degree level, n (%)	
Bachelor’s Degree	33 (38.4%)
Master’s Degree	26 (30.2%)
Doctorate	25 (29.1%)
Other ^a^	2 (2.3%)
**Clinical Characteristics**	
Mental Health Problem previously diagnosed (yes), n (%)	32 (37%) ^b^
ADHD	1 (1.2%)
Anxiety Disorder	2 (2.3%)
Bipolar Disorder	1 (1.2%)
Depersonalization–Derealisation Disorder	1 (1.2%)
Depression	9 (10.5%)
Eating Disorder	4 (4.7%)
PTSD	1 (1.2%)
Co-morbid Disorders	13 (15.1%)
Currently experiencing Mental Health Problems (yes), n (%)	22 (26%) ^c^
ADHD	1 (1.2%)
Anxiety Disorder	1 (1.2%)
Bipolar Disorder	1 (1.2%)
Co-morbid Disorders	12 (14%)
Depersonalization–Derealisation Disorder	1 (1.2%)
Depression	5 (5.8%)
Eating Disorder	1 (1.2%)
Mental Health Interventions, n (%)	28 (33%) ^c^
Talking Therapy	13 (15.1%)
Medication	10 (11.6%)
Talking therapy and Medication	5 (5.8%)
**Previous experience meditating and expectancy**	
Previous Experience Meditating, n (%)	45 (52%) ^d^
Mobile App	14 (16.3%)
Partial or completed attendance on a Mindfulness Based Program	9 (10.5%)
Mindfulness Retreat	3 (3.5%)
Other ^e^	16 (18.6%)
Not specified	3 (3.5%)
Expectations of benefits from the course (range: 0–10), Mn (SD)	7.09 (1.64)

^a^ e.g., PGCE. ^b^ Prefer not to say: n = 1; ^c^ Prefer not to say: n = 2; ^d^ Missing data: n = 11; ^e^ e.g., Yoga.

**Table 2 ijerph-18-06023-t002:** Complete cases analysis of primary outcomes.

Variable	Time	n	Mean (SD)	d	B (95% CI)	*p*
WEMWBS	T0	86	20.57 (3.61)			
	T1	74	22.76 (4.04)	0.59	2.08 (1.26 to 2.89)	<0.001
	T2	60	22.94 (4.09)	0.50	2.09 (1.22 to 2.97)	<0.001
CORE-10	T0	86	12.14 (6.75)			
	T1	74	9.11 (5.45)	−0.45	−2.63 (−3.91 to −1.36)	<0.001
	T2	60	9.15 (5.61)	−0.36	−2.39 (−3.77 to −1.01)	0.001

WEMWBS: Warwick Edinburgh Mental Wellbeing scale; CORE-10: Clinical Outcomes Routine Evaluation-10. d: Cohen’s d effect size from adjusted means. B: unstandardised regression coefficient using mixed models with subjects as random effects. 95% CI: 95% confidence interval.

**Table 3 ijerph-18-06023-t003:** The interaction effect between the time and mental health problems (current or previous diagnoses) on distress (CORE-10) scores using complete cases.

Mean (SD)	n	Mean (SD)	n	Time	d	B (95% CI)	*p*
Current case		Non-case					
17.30 (6.61)	23	9.98 (5.64)	61	T0			
11.13 (7.10)	16	8.23 (4.48)	56	T1	−0.99	−4.97 (−7.93 to −2.01)	0.001
11.09 (6.96)	11	8.32 (4.96)	47	T2	−0.89	−4.40 (−7.75 to −1.04)	0.010
Previous case		Non-case					
14.81 (7.58)	32	10.34 (5.53)	53	T0			
9.36 (6.47)	25	8.69 (4.52)	48	T1	−0.67	−3.94 (−6.58 to −1.30)	0.003
9.15 (6.04)	20	8.87 (5.23)	39	T2	−0.58	−3.41 (−6.27 to −0.55)	0.019

Current case: currently experiencing mental health problems. Previous case: previous diagnoses of mental health problems. d: Cohen’s d effect size from adjusted means. B: unstandardised regression coefficient using mixed models with subjects as random effects. 95% CI: 95% confidence interval.

**Table 4 ijerph-18-06023-t004:** Complete cases analysis on the proposed mediators.

Variable	Time	n	Mean (SD)	d	B (95% CI)	*p*
FFMQ-SF	T0	86	35.28 (7.06)			
	T1	74	40.14 (6.42)	0.62	4.45 (2.92 to 5.98)	<0.001
	T2	60	41.22 (6.38)	0.72	5.40 (3.75 to 7.05)	<0.001
SCS-SF	T0	86	2.67 (0.71)			
	T1	74	3.09 (0.69)	0.59	0.37 (0.24 to 0.49)	<0.001
	T2	60	3.18 (0.72)	0.67	0.41 (0.28 to 0.55)	<0.001
CDRISC	T0	86	24.76 (7.09)			
	T1	74	27.22 (6.58)	0.44	2.25 (1.10 to 3.41)	<0.001
	T2	60	28.07 (6.82)	0.49	3.03 (1.78 to 4.28)	<0.001

FFMQ-SF: Five-Facet Mindfulness Questionnaire-Short Form. SCS-SF: Self-Compassion Scale-Short Form. CDRISC: Connor–Davidson Resilience Scale. d: Cohen’s d effect size from adjusted means. B: unstandardised regression coefficient using mixed models with subjects as random effects. 95% CI: 95% confidence interval.

**Table 5 ijerph-18-06023-t005:** Complete cases analysis of student’s orientation towards their academic goals.

Variable	Time	n	Mean (SD)	d	B (95% CI)	*p*
Likelihood of achieving goal	T0	86	6.79 (1.56)			
	T1	74	7.20 (1.58)	0.28	0.40 (0.02 to 0.78)	0.039
	T2	60	7.42 (1.56)	0.34	0.57 (0.16 to 0.97)	0.006
Skills and resources	T0	86	7.12 (1.75)			
	T1	74	7.61 (1.66)	0.33	0.48 (0.12 to 0.85)	0.011
	T2	60	7.88 (1.59)	0.37	0.68 (0.28 to 1.08)	0.001
Commitment	T0	86	7.70 (1.74)			
	T1	74	8.49 (1.74)	0.43	0.74 (0.36 to 1.11)	<0.001
	T2	60	8.62 (1.49)	0.49	0.77 (0.37 to 1.17)	<0.001
Intrinsic motivation	T0	86	7.56 (1.91)			
	T1	74	7.58 (1.82)	0.01	0.01 (−0.41 to 0.44)	0.954
	T2	60	8.03 (1.66)	0.23	0.42 (−0.04 to 0.88)	0.075
Extrinsic motivation	T0	86	2.37 (2.41)			
	T1	74	2.88 (2.54)	0.21	0.50 (−0.08 to 1.07)	0.089
	T2	60	2.68 (2.05)	0.16	0.36 (−0.25 to 0.98)	0.249

Analyses of secondary outcomes. d: Cohen’s d effect size from adjusted means. B: unstandardised regression coefficient using mixed models with subjects as random effects. 95% CI: 95% confidence interval.

**Table 6 ijerph-18-06023-t006:** The mediating role of mindfulness on main outcomes.

		Direct Effects				Indirect Effects			
Outcome	R^2^	Path	Coef.	SE	95% CI †	Path	Boot.	SE	95% CI ‡
WEMWBS	0.27 ***	a	3.80 ***	0.98	1.84 to 5.76	a_1_*b_1_	1.10	0.43	0.38 to 2.05
		b	0.29 ***	0.06	0.16 to 0.42				
		c	1.96 ***	0.55	0.86 to 3.05				
		c’	0.86	0.54	−0.22 to 1.93				
CORE-10	0.27 ***	a	3.80 ***	0.98	1.84 to 5.76	a_1_*b_1_	−1.59	0.50	−2.65 to −0.69
		b	−0.42 ***	0.10	−0.62 to −0.22				
		c	−2.12 *	0.85	−3.82 to −0.41				
		c’	−0.53	0.83	−0.20 to 1.14				

WEMWBS: Warwick Edinburgh Mental Wellbeing scale. CORE−10: Clinical Outcomes Routine Evaluation-10. R^2^: multiple determination coefficient as an effect size measure. Coef: unstandardised regression coefficient. Boot: bootstrapped unstandardised regression coefficient. SE: standard error. 95% CI († 95% confidence interval; ‡ 95% bias-corrected bootstrap confidence interval for the indirect effect using 10,000 samples). “a*b” = indirect effects through mindfulness (see Figure 2). Path “c” refers to the unadjusted direct effects of X on Y. *** *p* < 0.001. * *p* < 0.05.

**Table 7 ijerph-18-06023-t007:** The mediating role of self-compassion on main outcomes.

		Direct Effects				Indirect Effects			
Outcome	R^2^	Path	Coef.	SE	95% CI †	Path	Boot	SE	95% CI ‡
WEMWBS	0.13 *	a	0.38 ***	0.08	0.23 to 0.54	a*b	0.98	0.45	0.22 to 1.98
		b	2.57 **	0.89	0.79 to 4.35				
		c	1.96 ***	0.55	0.86 to 3.05				
		c’	0.97	0.62	−0.27 to 2.21				
CORE−10	0.06	a	0.38 ***	0.08	0.23 to 0.54	a*b	−0.94	0.57	−2.13 to 0.11
		b	−2.44	1.44	−5.33 to 0.44				
		c	−2.12 *	0.85	−3.82 to −0.41				
		c’	−1.18	1.01	−3.20 to 0.84				

WEMWBS: Warwick Edinburgh Mental Wellbeing scale. CORE-10: Clinical Outcomes Routine Evaluation-10. R2: multiple determination coefficient as an effect size measure. Coef: unstandardised regression coefficient. Boot: bootstrapped unstandardised regression coefficient. SE: standard error. 95% CI († 95% confidence interval; ‡ 95% bias-corrected bootstrap confidence interval for the indirect effect using 10,000 samples). “a*b” = indirect effects through mindfulness (see Figure 2). Path “c” refers to the unadjusted direct effects of X on Y. *** *p* < 0.001. ** *p* < 0.01. * *p* < 0.05.

**Table 8 ijerph-18-06023-t008:** The mediating role of resilience on main outcomes.

		Direct Effects				Indirect Effects			
Outcome	R^2^	Path	Coef.	SE	95% CI †	Path	Boot	SE	95% CI ‡
WEMWBS	0.10 *	a	2.07 **	0.70	0.68 to 3.46	a*b	0.53	0.25	0.12 to 1.10
		b	0.25 *	0.10	0.06 to 0.45				
		c	1.96 ***	0.55	0.86 to 3.05				
		c’	1.43 *	0.57	0.30 to 2.56				
CORE-10	0.13 *	a	2.07 **	0.70	0.68 to 3.46	a*b	−0.69	0.40	−1.59 to −0.07
		b	−0.33 *	0.15	−0.64 to −0.03				
		c	−2.12 *	0.85	−3.82 to −0.41				
		c’	−1.43	0.87	−3.17 to 0.31				

WEMWBS: Warwick Edinburgh Mental Wellbeing scale. CORE-10: Clinical Outcomes Routine Evaluation-10. R2: multiple determination coefficient as an effect size measure. Coef: unstandardised regression coefficient. Boot: bootstrapped unstandardised regression coefficient. SE: standard error. 95% CI († 95% confidence interval; ‡ 95% bias-corrected bootstrap confidence interval for the indirect effect using 10,000 samples). “a*b” = indirect effects through mindfulness (see Figure 2). Path “c” refers to the unadjusted direct effects of X on Y. *** *p* < 0.001. ** *p* < 0.01. * *p* < 0.05.

## Data Availability

Following the International Committee of Medical Journal Editors (ICMJE), all of the individual anonymized and completely de-identified participant data are available for any analytical purpose that is related to achieve aims in the present study upon reasonable request to researchers (a) who provide a methodologically sound proposal and (b) whose proposed use of the data has been approved by an independent ethical review committee. The data and codebook will be provided by the corresponding author (willem.kuyken@psych.ox.ac.uk) to interested researchers that meet the aforementioned criteria.

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
