# Peer review of "The Mental Health and Wellbeing of University Students: Acceptability, Effectiveness, and Mechanisms of a Mindfulness-Based Course"

_ijerph, 2021, doi:10.3390/ijerph18116023_

Round 1

Reviewer 1 Report

Overall this is strong, well-written, and clear manuscript about the benefits of a mindfulness course for university students – an important topic to explore given the high rates of mental health issues that arise during university years.

Introduction

Page 1, line 45 - In the first sentence you say: A growing proportion of the population are benefitting from the social, occupational, 45 and academic opportunities offered by higher education

Tell me more what this means? How is the population growing – is it growing among typical college age students, or certain populations? Give us more details about what you are saying here.

Page 2, line 57 – It would be helpful to give some examples of how mental health issues impact quality of life – what aspects of quality of life

Page 2 line 58 – what does a “mentally healthy university” look like – include two or three examples

Materials and Methods

Page 5, line 209 – Here is says there were 10 groups, and I get that there 10 groups that did the 8 week program, but it took me a few reads to get there – I might try to make this more clear.

Page 8, table 1 – I believe gender should be on the line below where it currently is

Discussion

On page 13, line 473 you mention that home-practice was significantly related to pre-post invention improvement in distress and resilience – what are the implication of this? How could universities use this information?

Limitations

In the discussion it is mentioned that ethnicity predicted completion and that raises questions of inclusivity. Given pressures and societal contexts, especially when talking about mental health, it seems important to think about the way marginalized identities (rather racial, gender, gender identity, sexual orientation, migrant, etc) may play in role in the outcomes you examined. I understand this would be a large undertaking – and may not even be possible given the smaller sample size – but something needs to be added to your limitations that gets at this lack of investigation. These findings may be differential based on students identity.

Author Response

Thank you very much for the reviewers’ comments and suggestions. We have addressed all the queries in the revised version of the manuscript. Responses are below

Reviewer 2 Report

Dear Authors, 

Thank you for the opportunity to review the paper entitled “The mental health and wellbeing of university students: acceptability, effectiveness and mechanisms of a mindfulness-based course”. The contents of the manuscript are very interesting. The paper has a clear purpose, the clarity of the manuscript is high, with good scientific soundness. The manuscript contains an extensive introduction to the topic, as well as a solid discussion of the results obtained. Regrettably, it was not possible to provide an adequate sample size for the project. As we know the demand for interventions in the current pandemic situation is huge, is the project continued? I have no methodological objections, only two minor concerns:

  • I couldn't find the information, or misread when the study was conducted? It was indicated that interest was noted between October 2017 and April 2019. Does this mean that participants were consecutively recruited during this time?
  • I would recommend moving Tables 7 and 8 to the results section

Author Response

Thank you very much for the reviewers’ comments and suggestions. We have addressed all the queries in the revised version of the manuscript. Responses are below.
